# The Paradox of Ribosomal Insufficiency Coupled with Increased Cancer: Shifting the Perspective from the Cancer Cell to the Microenvironment

**DOI:** 10.3390/cancers16132392

**Published:** 2024-06-28

**Authors:** Giacomo D’Andrea, Giorgia Deroma, Annarita Miluzio, Stefano Biffo

**Affiliations:** 1National Institute of Molecular Genetics, INGM Fondazione Romeo ed Enrica Invernizzi, 20122 Milan, Italy; dandrea@ingm.org (G.D.); deroma@ingm.org (G.D.); miluzio@ingm.org (A.M.); 2Department of Biosciences, University of Milan, 20133 Milan, Italy

**Keywords:** SBDS protein, acute myeloid leukemia, neutropenia, Diamond–Blackfan anemia, CD4+, CD8+, immunoediting, immunotherapy, eIF6

## Abstract

**Simple Summary:**

Ribosomes are essential for the life of all cells. A reduction in the production of ribosomes always leads to a delay in cell cycle progression and to impaired growth, at least at the cellular level. We define ribosomopathies as a number of inherited monogenic diseases characterized by the partial loss of ribosomal factors. Not surprisingly, ribosomopathies show multi-organ signs and reduced cellular fitness. Strikingly, cancer is also a common comorbidity of ribosomopathies. The reconciliation of reduced growth with increased cancer risk poses an interpretative challenge. However, if we consider cancer as a systemic disease in which tumor cells thrive in a favorable microenvironment, we may find the right answers.

**Abstract:**

Ribosomopathies are defined as inherited diseases in which ribosomal factors are mutated. In general, they present multiorgan symptoms. In spite of the fact that in cellular models, ribosomal insufficiency leads to a reduced rate of oncogenic transformation, patients affected by ribosomopathies present a paradoxical increase in cancer incidence. Several hypotheses that explain this paradox have been formulated, mostly on the assumption that altered ribosomes in a stem cell induce compensatory changes that lead to a cancer cell. For instance, the lack of a specific ribosomal protein can lead to the generation of an abnormal ribosome, an oncoribosome, that itself leads to altered translation and increased tumorigenesis. Alternatively, the presence of ribosomal stress may induce compensatory proliferation that in turns selects the loss of tumor suppressors such as p53. However, modern views on cancer have shifted the focus from the cancer cell to the tumor microenvironment. In particular, it is evident that human lymphocytes are able to eliminate mutant cells and contribute to the maintenance of cancer-free tissues. Indeed, many tumors develop in conditions of reduced immune surveillance. In this review, we summarize the current evidence and attempt to explain cancer and ribosomopathies from the perspective of the microenvironment.

## 1. Introduction

One of the most puzzling aspects of cancer is related to ribosomopathies. The term ribosomopathy, of unclear origin, describes a clinical condition derived from the insufficiency of ribosomal proteins or of ribosome-associated factors. Ribosomes are essential components of all cells, and their depletion, in vitro, invariantly causes a loss of cell growth [1]. Clinical features associated with ribosomopathies include bone marrow failure, developmental abnormalities, and multi organ signs, as expected, in view of the ubiquitous nature of ribosomes. However, ribosomopathies also present an increased risk of cancer [2]. The increased risk of cancer, linked to a defect in cell growth and proliferation is paradoxical and has spurred an intense scientific debate. In this review, we will discuss the current state of the art. We will also introduce evidence suggesting that the paradoxical increase of cancer observed in patients with defects in the ribosomal machinery can be explained if we consider cancer as a disease in which the microenvironment plays a leading role. In particular, immunodeficiency is a well-known risk factor for cancer development.

## 2. Ribosome Biogenesis and Translation

Ribosome biogenesis [3,4,5] and translation [6,7] are described in several outstanding reviews. Hereafter, we will summarize the general framework. Ribosome biogenesis is a multistep process that involves ribosomal DNA (rDNA) transcription, ribosomal RNA (rRNA) processing, and the assembly of rRNAs with ribosomal proteins. Ribosome biogenesis occurs in the nucleolus. The nucleolus is a membraneless organelle that assembles around rDNA loci that encode for the 47S precursor rRNA (pre-rRNA), which is processed by endo- and exonucleases to give rise to mature 18S, 5.8S, and 28S rRNAs. 5S rRNA is independently transcribed. In the nucleolus, the two ribosomal subunits, known as 60S and 40S, are generated in a 1:1 ratio and then exported to the cytosol. Errors in any of the steps involved in ribosome biogenesis can cause nucleolar stress [8], which is characterized by alterations in nucleolar morphology and function. Nuclear transcription also produces tRNA molecules that are transferred to the cytosol [9]. 

It is widely established that translation is a cytoplasmic process, although controversial reports for nuclear translation exist [10]. Translation is divided into four steps, initiation, elongation, termination, and ribosome recycling. At initiation, eukaryotic initiation factors assemble with mRNA and 40S ribosomes, and the mRNA is scanned to the start codon. Initiation is rate-limiting for an individual mRNA. Although the start codon AUG has been unequivocally identified, several studies indicate that alternative codons such as CUG can initiate if in an optimal context [11]. Canonical cap-dependent translation is accompanied by other alternative ways of initiation, reviewed elsewhere [12]. Initiation is followed by elongation, characterized by 60S recruitment and peptide elongation. At stop codons, the ribosomes detach from the mRNA, and the protein is released. Ribosome recycling can be defined as the phenomenon by which recycling of ribosomes on the same mRNA is a different phenomenon from initiation ex novo. All these processes are energy consuming and regulated by signaling pathways that affect both initiation and elongation. Signaling to the translational machinery is described elsewhere [13]. In short, growth factors and nutrients stimulate translation and growth, whereas stressors inhibit translation and activate various savage pathways. In conclusion, ribosome biogenesis and translation are massive cellular processes that require the coordinated action of hundreds of genes under the control of signaling cascades. In this context, loss of ribosomes becomes pathogenic since it reduces the translation potential of cells. 

## 3. Ribosomopathies and Cancer at First Sight: A Simplified View

Several reviews have been published on the general scenario of ribosomopathies, and we refer to them for details [14,15,16]. For the sake of brevity, we will focus here on a recap of ribosomes in the context of ribosomopathies. Figure 1 shows a diagram of the process of ribosome biogenesis. Ribosomes are composed of two subunits, 60S and 40S. Each subunit is made by the assembly of ribosomal RNA with around 80 ribosomal proteins commonly known as rPS_n_ or rPLn, depending on their association with 40S (small) or 60S (large), respectively. (Of note, biochemists proposed a unified nomenclature for ribosomal proteins that is not centered on the human situation, but it does not match the human genome databases [17] and will not be used here). Since ribosomes are essential machineries for protein synthesis, the discovery that mutations in a ribosomal gene caused a non-lethal disease came as a surprise. The landmark discovery was that the partial loss of rPS19 caused a hereditary form of anemia known as Diamond–Blackfan Syndrome [18]. This very clear ribosomopathy was followed by other cases in which the partial loss of either ribosomal proteins or ribosome-associated factors caused inherited diseases. In general, haploinsufficiency is sufficient to cause disease, whereas total loss is incompatible with life. Overall, diseases caused by the insufficiency of ribosomal proteins/factors were named ribosomopathies. Well-defined ribosomopathies include Diamond–Blackfan anemia (DBA) caused by mutations in any one of several ribosomal proteins [19]; Treacher Collins syndrome (TCS), which is caused by mutations in genes that are involved in the transcription of ribosomal RNA [20,21]; Schwachman–Diamond syndrome (SDS), which is caused by mutations in the SBDS gene and other genes necessary for 60S maturation [22]; and 5q-syndrome due to rpS14 loss [23,24]. Other diseases are classified as ribosomopathies, although the function of the culprit gene may not be limited to ribosome biology. One interesting case is represented by congenital dyskeratosis, caused by mutations in the DKC1 gene. DKC1 deficiency leads to both a ribosomopathy-like disease due to the role of DCK1 in rRNA modification [25] and also to telomere shortening and consequent clinical manifestations [26]. 

Overall, all ribosomopathies present two puzzling issues. First, in spite of being due to the loss of ubiquitous proteins, they have tissue-specific alterations. These observations have generated research addressing the possibility that either (a) ribosome heterogeneity among tissues may explain tissue specific defects or (b) the threshold for ribosomal insufficiency may be different in different tissues [15]. Second, although several defects of ribosomopathies can be explained by reduced translation and reduced growth, patients affected by DBA or SDS have an higher risk for tumorigenic events [27]. Overall, increased cancer risk in the presence of impaired protein synthesis is paradoxical, especially because increased translation is a robust hallmark of cancer cells [28]. This paradox has been explained in several ways but mainly focusing on cancer as a cell-autonomous disease [16,27]. In short, the first step toward solving this paradox is to fully understand how ribosome biology impacts cancer.

## 4. Abundant Ribosomes and Translation Factors Favor Cancer Development in a Cell-Autonomous Fashion

In general, cancer correlates with an increased translational capability of cancer cells. Ribosomes are synthesized in the nucleolus and exported to the cytoplasm [29]. The existence of the nucleolus is known since early microscopy observations. Notably, the nucleolus is the first discovered organelle that forms upon phase separation [30]. The size of the nucleolus is variable, in vivo. Since the beginning of the twentieth century, pathologists have focused on the nucleolus as a parameter for the diagnosis of tumor malignancy. Hypertrophied nucleoli are a hallmark of cancer cells, as thoroughly discussed in a recent review [5]. In addition, extensive works have shown that the nucleolar size is a valid prognostic parameter in multiple neoplastic diseases [31]. The nucleolus can be effectively visualized by the histochemical AgNOR staining. The AgNOR staining technology has not been intensely developed in the pathological field, possibly due to its technical complexity. Nevertheless, malignant cancer cells are characterized by hypertrophic nucleoli, a proxy for ribosome biogenesis. Furthermore, the detection of larger nucleoli by the Ag-NORs staining is consistently associated with poor survival [32]. Thus, the more ribosomes there are, the more cancer cells are aggressive.

Individual ribosomal proteins are abundantly expressed in cancer cells. Ribosomal proteins are evolutionarily conserved, and most of them are unstable if free, i.e., not bound to ribosomes [33]. Thus, at first instance, the biochemical abundance of ribosomal proteins is a proxy for the ribosome abundance, rather than for the overexpression of a single ribosomal protein. Overexpression of multiple ribosomal proteins and its correlation with tumor malignancy have been observed by a multitude of studies [34,35,36]. Importantly, the production of elements of the ribosomal machinery is promoted by the amplification and expression of some of the most lethal oncogenes, for instance by the ones belonging to the Myc family [37,38]. The need for high ribosomal abundance by oncogenic stimuli is also proven by genetic evidence. rPL24 is a ribosomal protein of the 60S ribosomes. Genetic deletion of one rpL24 allele results in the suppression of Myc oncogenic activity [39]. The last discovered ribosomal protein is RACK1 [40], whose unconventional name derives from its first description as a Receptor for Activated PKC [41]. RACK1 ribosomal protein is also uniquely stable in the absence of ribosomes [40], but RACK1 deletion in mice phenocopies the loss of ribosomal proteins [42], indicating that its levels are linked to ribosomal production. RACK1 overexpression and correlation with cancer progression has been demonstrated by multiple studies [43,44,45]. To conclude, these selective studies and many others not cited here suggest that first, ribosomal protein abundancy reflects ribosome abundancy, and second, oncogenic signals must harness the ribosomal machinery [35,46,47,48,49]. 

mRNAs are converted onto proteins by translation. The study of translational control sheds light on why ribosomal abundance is essential to drive cancer. A specific description of this important area of research is outside of the scope of this review. Multiple works have described the key-elements of translation in detail [12,50,51]. In short, (a) mRNA abundance does not predict protein abundance, indicating that mRNA expression is necessary but not sufficient for the expression of the cognate protein [52], and (b) translation factors are important in selecting the translation of specific mRNAs under the control of signaling cascades [13]. In simple terms, the number of cellular ribosomes is not sufficient to translate all mRNAs, and, depending on the physiological status, free mRNA can range from 25 to 75% [53]. It is in this scenario that several translation factors regulated by oncogenic pathways drive the specific translation of oncogenic mRNAs. Notable examples are the following: eIF6 is a trans-acting factor for 60S ribosome biogenesis that also acts as a rate-limiting cytoplasmic translation factor [54]. Deletion of one allele of eIF6 dramatically reduces lymphomagenesis [55] and the transition from NAFLD (non-alcoholic fatty liver disease) to hepatocellular carcinoma [56]. The major cap-binding protein eIF4E is rate-limiting for cap-dependent translation of mRNAs critical for tumorigenesis [57] by two separate pathways. The MAPK/ERK pathway converges on Mnk1 and Mnk2 kinases, upstream of eIF4E. eIF4E function is regulated by the phosphorylation of the conserved serine 209 by Mnk1 and Mnk2. [58]. Importantly, this phosphorylation induction is essential for metabolic reshaping and oncogenic transformation [59,60] but dispensable for normal development [61]. In addition, the oncogenic mTOR pathway is essential for the stimulation of eIF4E activity through the inhibition of eIF4E repressor proteins 4E-BPs [62]. Finally, elongation of translation is also an important contribution to tumorigenesis and tumor growth [63,64]. In conclusion, translation factors downstream of oncogenic signaling pathways require abundant ribosomes to sustain the oncogenic program. In this specific context, a reduction of the ribosomal capability does not seem compatible with increased cancer malignancy.

## 5. Loss of Ribosomes and Cancer: Do Alterations in Ribosomal Stoichiometry Lead to Oncoribosomes?

If ribosomes are rate limiting for cancer growth as described before, how is it possible that ribosomal loss in ribosomopathies leads to increased cancer? In this context, a number of hypotheses backed by experimental data have been presented. From a historical perspective, seminal evidence obtained in the zebrafish model indicated that zebrafish lines with reduced ribosomal proteins developed malignant peripheral nerve sheath tumors. Tumorigenic lines included proteins both of the small ribosome and of the larger ribosome [65], suggesting that the culprit was the loss of the ribosomes and not of singular proteins. One interesting hypothesis that may explain the increased formation of tumors in the presence of haploinsufficient ribosomal proteins is the existence of unique cancer ribosomes. Oncoribosomes can be defined as specific ribosomes in which the lack (or unbalance) of a specific riboprotein makes them able to drive aberrant translation [66]. More generally, the idea is part of a model that stresses the existence of physiological ribosomal heterogeneity, widely discussed in other reviews [67]. The key elements of ribosomal heterogeneity are the following: (a) the stoichiometry of ribosomal proteins in tissues is variable, hinting at the presence of regional differences; (b) physiological differences in ribosomal composition drive tissue specific expression. The evidence of the existence of variations in ribosomal composition is relatively solid. In a study on mouse embryonic stem cells (MEFs), quantification of core ribosomal proteins evidenced that 6 out of 15 were substoichiometric, with four of them present in 60–70% of polysomal ribosomes. These findings strongly suggest the presence of actively translating ribosomes that lack at least one core ribosomal protein [68]. Experimental demonstration of the physiological roles for these alternative ribosomes is more difficult to obtain. Nevertheless, two studies separated by many years have provided genetic evidence on the relevance of heterogenous ribosomes. In mouse embryos mutated for *Rpl38*, global protein synthesis remains unchanged, but the translation of a subset of Homeobox mRNAs was impaired, confirming the existence of either Rpl38+ or Rpl38− ribosomes that confer transcript-specific translational control [69]. More recently, studies reported a peculiar ribosome, termed RibosomeST, characterized by a specialized nascent polypeptide exit tunnel. This ribosome is composed of the germ-cell-specific protein RPL39L, a paralogue of the core ribosomal protein rpL39. This ribosome, named Ribosome^ST^, cotranslationally regulates the folding of a subset of germ-cell-specific proteins that are fundamental for sperm formation [70]. Overall, the value of these two studies and of the many others published in the meantime is to show that heterogenous ribosomes are not incompatible with life [67]. 

Provided that heterogenous ribosomes can exist in normal conditions, the possibility that oncoribosomes exist becomes real. The question is, however, whether the existence of oncoribosomes can be convincingly demonstrated. An oncoribosome must be demonstrated by a combination of genetic studies, biochemical studies, and preclinical evidence. Heterogeneity on ribosomal protein abundance can be observed in cancer cells [71,72], but is it tolerated or a driving event? Data are not conclusive. For instance, overexpression of rpL15 [35] is clearly associated with metastatic formation and resistance to chemotherapy, but mechanistically, it is unclear whether it leads to aberrant ribosomes or simply increases the capability of the ribosomal machinery, as described before. Thus, with the exception of a general amplification of ribosomal proteins, namely of the whole translational capability, the existence of a specific oncoribosome has never been demonstrated. Yet, the possibility cannot be dismissed. In a yeast model, the RPL10-R98S mutation found in T-cell acute lymphoblastic leukemia (T-ALL) increases stop-codon read-through and near-cognate amino acid misincorporation. This suggests that perhaps in some conditions, altered ribosomes are not a by-product of cancer but can become a driving force [73]. In humans, the evidence is still missing. 

## 6. Loss of Ribosomes and Cancer: Are Ribosomal Proteins Tumor Suppressors?

A second possibility that may explain increased cancer incidence when a ribosomal protein is mutated or reduced is due to a non-ribosomal function of a ribosomal protein. It has been recently proposed that several proteins may have secondary functions, e.g., moonlighting activity [74]. Moonlighting activity of proteins is fascinating and logical from an evolutionary perspective but obviously difficult to demonstrate by genetic studies in which only the primary function of a gene can emerge. One case is represented by RACK1 that in spite of being, unequivocally, a 40S-associated structural ribosomal protein [42,75], it has more than fifty described functions and perhaps can be better defined as an adaptor protein [76]. In this context, we expect that the loss of some ribosomal proteins is protumorigenic because they may also act as tumor suppressors. Specific differences appear immediately when comparing classic tumor suppressors to ribosomal proteins. Classic tumor suppressors are lost in a bi-allelic fashion [77]; however, in the case of ribosomal proteins, this is not normally possible since bi-allelic loss is lethal due to gene essentiality. In addition, the relationship between ribosomal proteins and tumor suppression is rather intricate. First, loss of ribosomes may induce tumor suppressor pathways. The most well-known tumor suppressor pathway is represented by tp53. The tp53 gene product encodes for a transcription factor, p53, which is a critical barrier to carcinogenesis. Inactivation of tp53 is the most common mutation in sporadic human cancers [78]. p53 is thought to act as a tumor suppressor by serving as a cellular stress sensor, its mutations are described in many reviews [79]. In unstressed cells, p53 is targeted by the E3 ubiquitin ligase MDM2 for degradation, keeping p53 at low levels. A variety of classical stress signals, including DNA damage, oncogene expression, and hypoxia, relieve p53 from MDM2 inhibition, thus inducing p53. In this context, ribosomal stress (Figure 2) can be defined as the induction of p53 following alterations of ribosomal biology [80]. Induction of ribosomal stress and of p53 is a common feature of ribosomopathies. It is observed in DBA [81], SDS [82], and experimental models caused by mutations of small nucleolar RNA (snoRNA) SNORD118 [83]. In a recent work, a systematic screening of human ribosomal proteins revealed that uL5 (RPL11 in the human genomic database) and uL18 (RPL5) were the strongest contributors to nucleolar structure maintenance and ribosomal stress [84]. The same study found that the p53 steady-state level increased from 0 to 10-fold upon depletion of ribosomal proteins and that a third of the ribosomal proteins were found to affect p53 level at least fivefold [84]. In short, the loss of ribosomal proteins induces a tumor suppression pathway rather than a loss of a tumor suppression pathway.

In contrast, several reports have identified specific genetic networks in which ribosomal proteins crosstalk with p53 at the molecular level and directly regulate its activation. During nucleolar stress, ribosomal biogenesis is disrupted, causing various ribosome-free ribosomal proteins to interact with MDM2. This binding displaces p53, allowing its stabilization and activation. Similarly, ribosomal proteins released from the breaking down of cytoplasmic ribosomes can interact with MDM2 [85]. Mechanistically, the following data have been reported. First, a complex formed by rpL5/rpL11/5S rRNA binds MDM2, alleviating its inhibitory effect over p53 [86]. Loss of tumor suppressor “RPL5/RPL11” does not trigger cell cycle arrest but hampers proliferation by reducing ribosome content and translation capacity [87]. Second, mutations in the ribosomal proteins rpL5 and rpL10 are found in 12 of 122 (9.8%) pediatric T-ALLs, with recurrent alterations specifically affecting Arg98 in rpL10 [88]. Together these data are consistent with a role for wt rpL5 in tumor suppression. In this context, we can hypothesize that losing a ribosomal protein may somehow weaken the tumor suppression. In conclusion, if ribosomal proteins act as tumor suppressors, their loss may pave the way to cancer. However, ribosomal loss seems to activate tumor suppression rather than impair it. In addition, ribosomal loss diminishes proliferation. The way that ribosomal protein deficiency increases cancer incidence in a cell-autonomous fashion remains, therefore, far from linear. A possibility is that prolonged ribosomal stress may induce compensatory changes of hyperproliferation. We should also explore plausible mechanisms that explain in a rationale way how the loss of a ribosomal protein can increase cancer in an organism.

## 7. Immunological Defects Driven by Translation and Cancer as the Result of a Poor Immunological Response

In this section, we will first briefly discuss how cancer cells develop in the absence of an efficient immune system and then define the impact of alterations of the translational machinery on T cells, a key cellular element of adaptive immunity. In one of the most famous reviews on cancer, tumorigenesis was described as a multistep process involving the acquisition of six biological features essential for the maintenance and survival of the tumor itself. These hallmarks serve as an organizing principle and are as follows: maintaining proliferative signaling; escaping growth suppressors; preventing cell death; enabling replicative immortality; stimulating angiogenesis; and activating invasion and metastasis. In later studies, however, a new hallmark appeared, the capability of tumor cells to evade the immunological response [89]. The process of immunoediting in tumors is discussed in a variety of reviews [90]. This concept has experienced a century of controversies [91], but it is now well established. Cancer immunoediting proceeds through three phases: elimination, equilibrium, and escape. In the first phase, elimination, neoantigens produced by tumor cells are recognized by the adaptive repertoire of CD4+ and CD8+ T cells. Subsequently, the cytotoxic action of CD8+ cells leads to the elimination of cancer cells. Slowly, the tumor adapts to the immune system and escapes it during the advancement of the disease. Several landmarks witness the role of the immune system in the elimination of cancer cells. We will list some of them. One, the HIV virus infects and kills CD4+ lymphocytes and progressively leads to the reduction of the counts of CD4+ cells to less than 200 per milliliter of blood. When the number of CD4+ cells drops, AIDS syndrome develops. AIDS claims as major symptoms both opportunistic infections and tumor formation [92]. Two, primary immunodeficiency diseases (PIDDs) are a group of over 300 distinct genetic defects that affect the immune system [93]. A consistent increase in lymphoma among both men (10-fold increase, *p* < 0.001) and women (8.34-fold increase, *p* < 0.001) diagnosed with PIDD has been reported [94]. Several other studies link immunodeficiency with cancer [95]. Third, reactivation of the immune system by immunotherapy represents the most successful novel paradigm of cancer therapy [96,97,98,99]. Importantly, the success of immunotherapy is associated with the abundance of novel mutations. In other terms, the more neoantigens a tumor has, the higher the chance for successful immunotherapy [100]. 

We will now discuss the fact that immune cells are particularly sensitive to the functionality of the translational apparatus. The immune response requires strong activation of the translational machinery (Figure 3). Already, seminal work from long ago has shown that in vivo intraperitoneal injection of Salmonella typhi led to an increase of activity of free and membrane-bound ribosomes of the spleen [101]. Levels of proteins crucial to both innate and adaptive immunity are regulated at the level of translation [102], with the strong contribution of the mTOR kinase pathway. Dendritic cells (DCs) are the sentinels of the mammalian immune system, characterized by antigen presentation to adaptive immunity cells. Besides immune-related biochemical pathways, a key biological function observed among translationally regulated mRNAs during dendritic cell maturation is protein synthesis itself [103]. CD8+ T cells play a pivotal role in controlling intracellular infections and immune responses against tumors. During an acute infection or the generation of neoantigens, antigen-specific CD8+ naive T cells rapidly increase in number and differentiate into CD8+ effector (Teff) cells. Teff cells subsequently eliminate the infected or mutated cells [104]. Antigen-specific T cells dynamically regulate gene expression at the translational level via the T cell antigen receptor (TCR) signaling. During the T cell clonal expansion phase, the translation of mRNAs encoding different components of translation machinery, including ribosomal proteins, is mandatorily upregulated [105]. In general, T cell activation fully requires an explosive expansion of the translational machinery through the eIF4E-eIF4EBP axis [106,107]. In its absence, the production of cytokines stimulating the immune response, such as interferons, are inhibited [108]. In adaptive immune response, activated mature B cells generate antibody-secreting plasma cells. If a genetic mutation or pharmacological inhibition disrupts the interaction between eIF4E and eIF4G, cap-dependent translation is compromised. This translational impairment within activated B cells results in a reduction of antibody class switching [109]. As described above, eIF6 activity regulates the rate of translation downstream of signaling cascades [40,54]. In human CD4+ T cells, eIF6 levels rapidly increase following T-cell receptor activation, promoting the glycolytic switch and the development of effector functions. Notably, in these cells, eIF6 levels regulate the secretion of interferon-γ (IFN-γ). [110]. Finally, tumor-infiltrating lymphocytes (TILs) are the key players of immunity within tumors. Protein synthesis rates measured through puromycin incorporation demonstrated an increase in translating CD4+ and CD8+ cells within tumors compared to normal tissues. [111]. In addition to a general requirement for protein synthesis during the immune response, deficiency of the specific translation of some mRNAs impairs the immune response. In the absence of MCTS1, a regulator of translational reinitiation, JAK2 translation in innate-like adaptive T lymphocytes is reduced, leading to impaired IL-23-dependent induction of IFN-γ [112]. In conclusion, (a) a decrease in the response of the immune system is a hallmark of cancer, and (b) translational activation of lymphocytes and other players of adaptive immunity, such as dendritic cells, is essential for the immune response. These, and many other data [102,113] indicate that the proper immunological response requires a massive mobilization of the translational apparatus. These data can be summarized as follows: The increase of cancer in an organism in which ribosomal proteins are impaired is not necessarily due to the fact that abnormal ribosomes lead to cancer by altering the prospective cancer cell. Rather, it is possible that a reduction in the efficiency of the immune response, or an alteration in the microenvironment, favors the emergence of weak, ribosome-deficient cancer cells. 

## 8. Bone Marrow Deficits and Tumors

Developmental studies identify cell-autonomous and non-cell-autonomous developmental decisions. In the first case, a cell evolves in the absence of external stimuli, and in the latter, it requires external stimuli [114]. The question that we will try to address now is whether cancer onset in ribosomopathies can also be caused by insufficient ribosome function in the host tissue rather than aberrant translation in the cancer cell. Schwachman–Diamond Syndrome (SDS) is a ribosomopathy characterized by increased risk for acute myeloid leukemia, among pancreatic insufficiency and growth deficits [115]. In the classic transformation assay, primary fibroblasts are transduced with oncogenic viruses, and the number of transformed colonies is measured. Oncogenic strength, namely the intrinsic predisposition to tumorigenic transformation, is scored by colony numbers [55]. Years ago, we recovered mouse embryonic fibroblasts from a SDS model and matched wt animals. We performed tumorigenic transformation of the fibroblasts. We consistently obtained around one-fifth the number of colonies in SDS compared to wt mice. In addition, the few tumors generated from SDS mice did not grow in xenografts, whereas wt tumors thrived [116]. These data unequivocally demonstrate that the ribosomopathy mutation in SDS is not oncogenic in a cell autonomous function but rather tumor suppressive. These data suggest that the cause of increased cancer in SDS patients is the alteration of the surrounding environment. If so, how does this happen? SDS symptoms include, in almost all individuals, neutropenia [117]. The occurrence of myelodysplastic syndrome (MDS) or acute myeloid leukemia (AML) in patients with congenital neutropenia is well-documented and represents a significant cause of mortality [118]. One possibility is that hematopoietic stressors may select hematopoietic stem cells carrying certain mutations, leading to their clonal expansion, due to the specific features of the microenvironment, rather than the cancer cell itself. In addition, in these conditions, immune cells may fail to properly eliminate weak mutant cells. Indeed, in neutropenia, the bone marrow microenvironment changes at several levels, and in some cases, tumor-initiating events are present in non-hematopoietic cells. For instance, genetic experiments showed that mice lacking nuclear factor-κB (NF-κB) inhibitor-α (IκBα), specifically in non-hematopoietic cells, developed myeloproliferative neoplasm in adjacent cells [119]. How an altered microenvironment may affect the development of tumors is widely discussed in some reviews [120].

Another interesting aspect of ribosomopathies is the pressure for clonal expansion of mutant cells that may have increased fitness compared to the original population. Clonal expansion can lead to protumorigenic events. There is an interesting case in SDS itself (Figure 4). Independent works have shown in SDS patients clonal expansion of two types of mutations. In some SDS patients, recurrent somatic eIF6 mutations occur in the myeloid lineage [121,122]. eIF6 is a ribosome and translation factor necessary for ribosome biogenesis and efficient translation [54]. These data impressively confirm the seminal observation that similar mutations, in the yeast model, partially rescue the loss of SBDS, the culprit gene of SDS [123]. Functional models from these studies suggest that EIF6 mutations lead to improved translational efficiency in patients with SDS, possibly through a partial loss of function of eIF6. Importantly, the prognosis of patients with eIF6 mutations is generally favorable, indicating again that the loss of ribosomal components, if anything, decreases cancer malignancy. In other SDS patients, mutations of the tumor suppressor *TP53* are enriched, suggesting that they are early initiating events that precede the development of a myeloid malignancy. In the case of tp53 mutations, the prognosis is negative [124]. We can reconcile these studies with the observation that SBDS-deficient cells are less prone to transformation, if we hypothesize that the increased rate of acute myeloid leukemia is due to prolonged cell stress, in a cell autonomous fashion, and leads to either maladaptive or adaptive mutations, respectively, for tp53 and eIF6.

The final issue is whether components related to immune surveillance are affected in ribosomopathies. Neutropenia conditions are accompanied by higher infection risk [125]. Research in the last decade has highlighted the crucial role of neutrophils in immune response, revealing the complex nature of these cells, as they not only perform effector functions in innate immunity but actively shape the adaptive immune response by directly interacting with dendritic cells and lymphocytes or indirectly influencing their activity through cytokine production [126].

CD4+ and CD8+ T cells in SDS are, in homeostatic conditions, normal [127]. However, inhibition of Sbds (SDS gene) is associated with a decrease in circulating B lymphocytes, despite evidence of normal B lymphopoiesis [128]. In short, current evidence shows signs of a minor impairment or no impairment of lymphocytic function in the blood of unchallenged SDS patients. This limited evidence does not exclude at all the possibility that increased tumor formation is due to the combination of ribosomal stress in a cell autonomous fashion and poor performance of immunosurveillance or alterations of the hematopoietic niche. In mice models, for instance, moderate pharmacological inhibition of eIF4E activity reduces CD4+ T cell differentiation to TFH cells, germinal center formation, and B cell maturation to antigen-specific plasma cells without showing changes in homeostatic conditions [129]. In humans, efficient translation drives the transition between CD4+ T cell subtypes [107]. As discussed before, the translational apparatus is required for the efficient immune response; thus, a minor defect in circulating B lymphocytes in homeostatic conditions is likely to imply a massive loss of function in conditions of immunological challenging and not the other way around.

Data from Diamond–Blackfan anemia also suggest impaired tumor immunosurveillance. In America, it has been reported that a quarter of DBA patients die by age 50 due to both cancerous and non-cancerous complications. It is now officially recognized that DBA patients have an increased susceptibility to developing malignancies. Among various reported solid tumors, the highest risks are for colon carcinoma and osteogenic sarcoma, tumors that do not arise in the context of the hematopoietic niche [130,131]. At the same time, more than half of patients show quantitative deficits in serum immunoglobulins and/or circulating T, natural killer, and B lymphocytes [132]. The clonogenic cell output per long-term initiating cultures is significantly lower in DBA patients [133]. Overall, results suggest that the underlying defect in patients with severe refractory DBA may not be limited to the erythroid lineage [134].

Finally, the therapeutic outcome of SDS patients reveals that tumorigenicity and lethality is not necessarily associated with the malignancy of tumor cells but with the effect of ribosomal mutations in the host microenvironment. Cytoreductive chemotherapy for leukemia in Schwachman-Diamond syndrome failed to prevent relapse and was unsuccessful due to unacceptably high toxicity [135]. These observations further suggest that it is not that the “ribosomopathy” cancer cell is particularly aggressive but rather that the microenvironment is permissive. Unfortunately, a non-invasive time-course analysis of the microenvironment, in situ, is not yet feasible.

## 9. Concluding Remarks and Limitations

Ribosomopathies present an increase in specific cancer types in the presence of reduced growth. This apparent paradox can be explained by understanding that cancer cells behave in a cell autonomous fashion in vitro, but in vivo, they can thrive only if they escape the immune system. Thus, it is conceivable that increased tumor formation in ribosomopathies is due to decreased cancer immunosurveillance, because lymphocytes require robust protein synthesis to fully reach effector properties. This hypothesis should be tested more vigorously. Future studies should aim at defining whether the immune system normally behaves in the presence of mutations of the ribosomal apparatus. Specifically, conditional mouse models with lymphocyte-specific haploinsufficiency of ribosomal proteins may solve the riddle. Bringing back the study of ribosomopathies, in vivo, in mouse models is essential to a better comprehension of the disease and may lead to better therapies. 

## Figures and Tables

**Figure 1 cancers-16-02392-f001:**
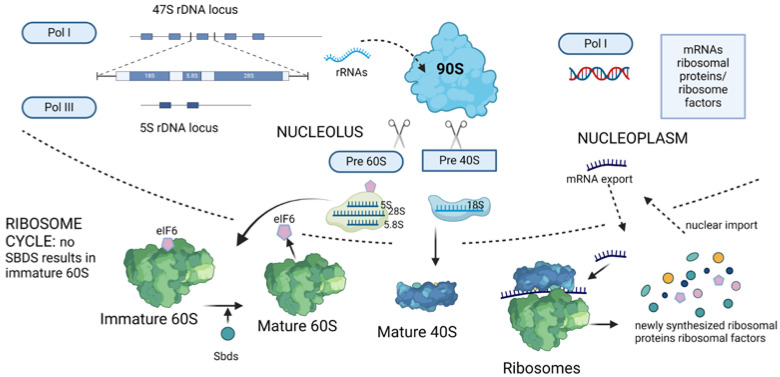
**Eukaryotic ribosome biogenesis.** Ribosome biogenesis is a stepwise flow of catalytic reactions that starting from the nucleolus leads to mature ribosomes in the cytoplasm. In the nucleolus, RNA polymerase I (Pol I) transcribes a polycistronic rRNA precursor that is assembled by ribosome biogenesis factors (RBFs) and ribosomal proteins (RPs) into the (precursor) pre-90S particle. A number of defined steps follow. In particular, endonucleolytic cleavage of the pre-90S particle gives rise to the pre-40S, which contains 18S rRNA, and to the pre-60S particle, which contains the 28S and 5.8S rRNA. In the nucleus, RNA polymerase III (Pol III) synthesizes the 5S rRNA that will then be assembled into the pre-60S particle. A mutation in any of the genes that intervene in ribosome biogenesis may generate a ribosomopathy. Here we show eIF6 and SBDS. eIF6 binds the pre-60S particle and impedes its premature joining to pre-40S. After quality control, 60S and 40S are exported to the cytoplasm, where the last maturation of ribosomes occurs. SBDS intervenes in 60S maturation by regulating eIF6 eviction. SBDS loss of function generates a ribosomopathy known as Schwachman–Diamond syndrome (SDS).

**Figure 2 cancers-16-02392-f002:**
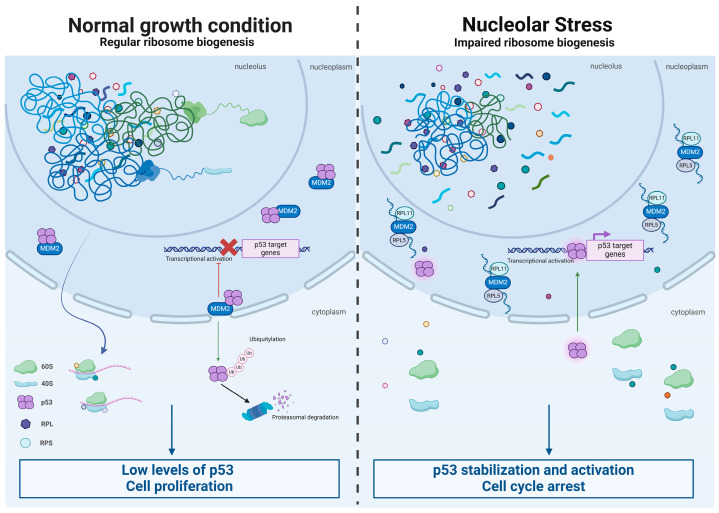
**Nucleolar stress driven by ribosomal loss activates tumor suppression.** Under normal growth conditions, small (40S) and large (60S) ribosomal proteins (RPs) are assembled in the nucleolus and transported to the cytoplasm for protein synthesis. When cells are healthy, p53 is targeted by the E3 ubiquitin ligase MDM2 for degradation, keeping p53 at low levels. During nucleolar stress, ribosome biogenesis is inhibited, and ribosome-free forms of ribosomal proteins (RPL and RPS) accumulate in the nucleoplasm. Specifically, rpL5-rpL11-5srRNA forms a ternary complex that interacts with MDM2. This event leads to the stabilization of p53, which in turn promotes the transcription of its downstream targets.

**Figure 3 cancers-16-02392-f003:**
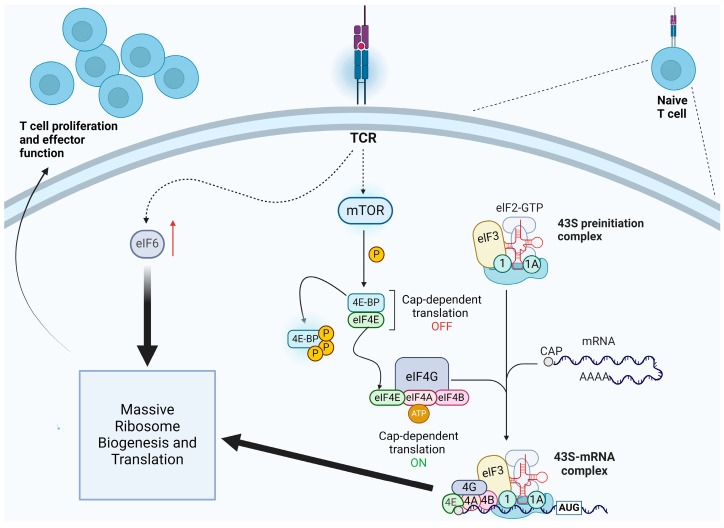
Cells of the immune system require dynamic regulation of translation for the acquisition of their effector functions. In activated T cells, during the clonal expansion phase, the translation of ribosomal protein (RP) mRNAs is significantly upregulated. In particular, upon TCR stimulation, activation of the mTOR pathway allows the phosphorylation of 4EBPs and their release from eIF4E, which can now join eIF4G to form the eIF4F-mRNA complex and allow cap-dependent initiation of translation. The eIF4E-4EBP axis triggers the amplification of the translational machinery and plays a pivotal role in the activation and differentiation of T cells. Enhanced eIF4E activity within T cells boosts the secretion of cytokines stimulating the immune response, such as IFNγ. Moreover, in human CD4+ T cells, eIF6 levels rapidly increase upon TCR activation and drive the glycolytic switch as well as increased ribosome biogenesis and acquisition of effector functions. In short, these and other data show that the immune response requires a massive translational upregulation.

**Figure 4 cancers-16-02392-f004:**
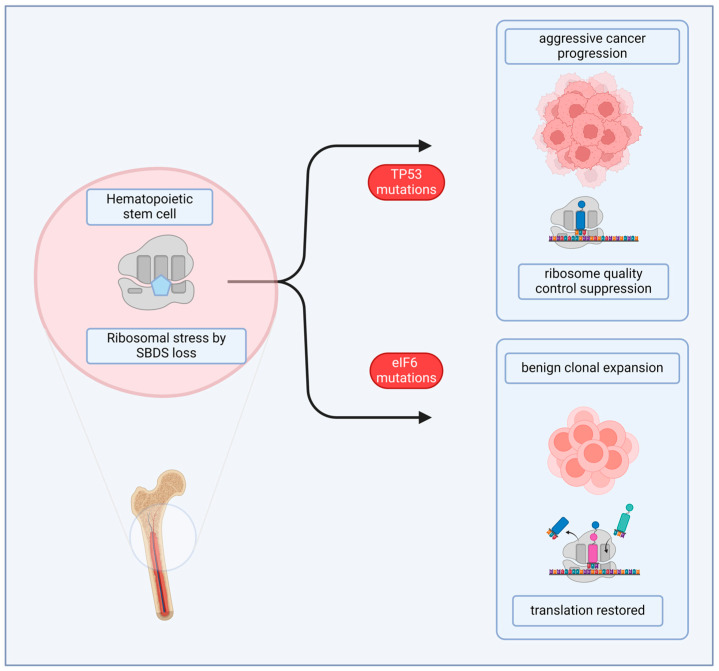
**Ribosomopathy evolution can be either adaptive or maladaptive.** Schwachman–Diamond syndrome (SDS) is caused by the partial loss of function of the Sbds protein and may evolve into cancer. Sbds may regulate eIF6 function on ribosomes. In SDS patients, abnormal expansion of hematopoietic cells can be found. Interestingly, some patients lose the p53 tumor suppressor, possibly overcoming ribosomal stress. This mutation, however, greatly predisposes to cancer. Other patients may have mutations in the eIF6 gene. In this case, altered eIF6 structure may have a compensatory effect on Sbds loss without facilitating cancer development.

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
