# Peer review of "The Paradox of Ribosomal Insufficiency Coupled with Increased Cancer: Shifting the Perspective from the Cancer Cell to the Microenvironment"

_cancers, 2024, doi:10.3390/cancers16132392_

Round 1
Reviewer 1 Report
Comments and Suggestions for Authors
It appears that this manuscript is a very excellent review about the paradox that exists in cases of ribosomopathies and the consequent increase in the occurrence of cancer. The introduction is very well written and introduces the reader to the essence of the issue in an easy way. They emphasize that changes in the structure of ribosomes can lead to so-called oncoribosomes, and the possibility that some ribosomal proteins may have moonlighting activity further highlights the enormous complexity of the process. Although there are already excellent review articles on ribosomopathies in the literature, this manuscript further considers the role of immune system surveillance, or rather the lack thereof, in ribosomopathic situations.
Author Response
General Rebuttal
We thank the reviewers for their time spent reading our work. Hereafter our responses.
Reviewer 1 has not expressed major criticism.
A: Thanks for the evaluation.
Reviewer 2 has expressed major criticism. In particular:
In this review, the author suggests that a defect in immune surveillance might be responsible for this paradox. While this is an interesting perspective, the review could benefit from more substantial evidence to support this hypothesis. It currently appears somewhat speculative. I agree that translation plays an important role in T and B cell function. However, there is limited evidence suggesting that the function of the adaptive immune system decreases in ribosomopathy patients. The authors cite Iskander et al. (ref 132) regarding immune defects in DBA patients; however, the study did not find a significant overall correlation between the RP gene mutated and the presence of immunodeficiency. Additionally, some patients in this cohort do not have ribosomal mutations.
Another compelling point is why ribosomopathy patients are predisposed to specific cancers. It seems that the transformation from a hypoproliferative to a hyperproliferative phenotype may be multifactorial. Is there any evidence for inflammation or metabolism dysregulation playing a role?
If there is not sufficient evidence of an immunosurveillance defect in ribosomopathy patients, it would be helpful to clearly state in their review that this is a speculative hypothesis. This would provide a more balanced and accurate presentation of the current understanding in this field.
A: We thank the reviewer for the comments. I believe that there is a long discussion to be done here. In general, I agree with the point that there is no conclusive evidence for the role of the immune system in the formation of tumors in patients affected by ribosomopathies. However, the alternative explanations that do not take in account the immunodeficiency theory and have been put forward have even less evidence and are, at least to me, even more disturbing. At the very least the connection between any form of immunodeficiency and tumor onset and progression is known since fifty years (Kersey et al., 1973), whereas theories of oncoribosomes/lack of (ribosomal) tumor suppression do not reach the minimal threshold for being conclusive anywhere. The same seminal zebrafish paper of Amsterdam et al.(Amsterdam et al., 2004) has not been followed by experiments aiming at understanding whether the increased tumor burden in a specific cell type was cell autonomous or not. This said we have incorporated the criticism in the conclusions, toning down our language and defining the need for more conclusive studies (line 500 and below of the new version).
Please allow me to further elaborate. To define “speculative” the immunodeficiency theory is not correct. I list here two observations on SDS that are enlightening: SBDS-deficient cells, in vitro, are poorly transformed indicating the lack of inherent instability (Calamita et al., 2017). Twenty years ago, the standard of care for SDS patients was perhaps more limited, and it was reported that children were harmed by infections (Dror et al., 2001). The Iskander paper shows measurable differences in up to half patients in unchallenged conditions (Iskander et al., 2019), it is true that some patients had not rp mutations, but they were not a real control group. The presence of an observable, even minimal, difference in unchallenged conditions is a strong, predictive sign of immunodeficiency in challenged conditions, as shown by tens of mouse models that have a phenotype only upon challenging, see for instance (Manfrini et al., 2017) where 15% of mice succumb to norovirus infection. I hope that I convinced you that the question on immunodeficiency is not at all speculative.
I perfectly agree on your point on the microenvironment. The change from a hypoproliferative to a hyperproliferative disorder is certainly multifactorial. The raised question is very important. At the cellular level, it is always positive selection for fast growers. Here, we have a technical problem, however. The description of immunological parameters is feasible on blood and can rely on a multitude of tools and studies. Vice versa, we lack a capability to quantitatively and timely discuss the (bone) microenvironment, for obvious technical reasons. Statement included. Thank you very much for your attention.
Reviewer 3 has not expressed major criticism.
A: Thanks for the evaluation.
Amsterdam, A., Sadler, K.C., Lai, K., Farrington, S., Bronson, R.T., Lees, J.A., and Hopkins, N. (2004). Many ribosomal protein genes are cancer genes in zebrafish. PLoS Biol 2, E139.
Calamita, P., Miluzio, A., Russo, A., Pesce, E., Ricciardi, S., Khanim, F., Cheroni, C., Alfieri, R., Mancino, M., and Gorrini, C. (2017). SBDS-deficient cells have an altered homeostatic equilibrium due to translational inefficiency which explains their reduced fitness and provides a logical framework for intervention. PLoS genetics 13, e1006552.
Dror, Y., Ginzberg, H., Dalal, I., Cherepanov, V., Downey, G., Durie, P., Roifman, C.M., and Freedman, M.H. (2001). Immune function in patients with Shwachman–Diamond syndrome. British journal of haematology 114, 712-717.
Iskander, D., Roberts, I., Rees, C., Szydlo, R., Alikian, M., Neale, M., Harrington, Y., Kelleher, P., Karadimitris, A., and de la Fuente, J. (2019). Impaired cellular and humoral immunity is a feature of Diamond‐Blackfan anaemia; experience of 107 unselected cases in the United Kingdom. British Journal of Haematology 186, 321-326.
Kersey, J.H., Spector, B.D., and Good, R.A. (1973). Primary immunodeficiency diseases and cancer: The immunodeficiency‐cancer registry. International journal of cancer 12, 333-347.
Manfrini, N., Ricciardi, S., Miluzio, A., Fedeli, M., Scagliola, A., Gallo, S., Brina, D., Adler, T., Busch, D.H., and Gailus-Durner, V. (2017). High levels of eukaryotic Initiation Factor 6 (eIF6) are required for immune system homeostasis and for steering the glycolytic flux of TCR-stimulated CD4+ T cells in both mice and humans. Developmental & Comparative Immunology 77, 69-76.
Reviewer 2 Report
Comments and Suggestions for Authors
Ribosomopathies are intriguing disorders. Mutations in ribosomal proteins or factors involved in translation lead to ribosomopathies. One of the important questions in this field is, ribosomal proteins are ubiquitously present, then why do mutations in these proteins cause tissue-specific phenotypes rather than global phenotypes? Several models such as ribosome heterogeneity, extra-ribosomal functions of ribosomes, the ribosome concentration model, and p53-mediated cell death have been proposed.
Another interesting observation is that some ribosomopathy patients, but not all, develop cancer. This presents a paradox, how does a hypoproliferative phenotype become hyperproliferative?
In this review, the author suggests that a defect in immune surveillance might be responsible for this paradox. While this is an interesting perspective, the review could benefit from more substantial evidence to support this hypothesis. It currently appears somewhat speculative. I agree that translation plays an important role in T and B cell function. However, there is limited evidence suggesting that the function of the adaptive immune system decreases in ribosomopathy patients. The authors cite Iskander et al. (ref 132) regarding immune defects in DBA patients; however, the study did not find a significant overall correlation between the RP gene mutated and the presence of immunodeficiency. Additionally, some patients in this cohort do not have ribosomal mutations.
Another compelling point is why ribosomopathy patients are predisposed to specific cancers. It seems that the transformation from a hypoproliferative to a hyperproliferative phenotype may be multifactorial. Is there any evidence for inflammation or metabolism dysregulation playing a role?
If there is not sufficient evidence of an immunosurveillance defect in ribosomopathy patients, it would be helpful to clearly state in their review that this is a speculative hypothesis. This would provide a more balanced and accurate presentation of the current understanding in this field.
Author Response

(The authors gave the same response as above.)

Reviewer 3 Report
Comments and Suggestions for Authors
The manuscript by D‘ Andrea et al., entitled “ The paradox of ribosomal insufficiency coupled with increased cancer: shifting the perspective from the cancer cell to the microenvironment “
Provide interesting information about ribosomopathies and their potential as predictive markers for cancer risk. The manuscript is well written, and the presentation of diagrams is in the best form and supports the context of the review. This manuscript provides very interesting information for the scientific community. This work opens new chapter for researchers, who are interesting evaluate the potential of ribosomopathies in cancer diagnostic and therapy. This manuscript will be interesting for the reader of the journal. I suggest publishing this manuscript in the present form
Many thanks
Author Response

(The authors gave the same response as above.)
